# Handover Decision-Making Algorithm for 5G Heterogeneous Networks

**Mark Irwin Goh** [1], **Abbas Ibrahim Mbulwa** [1], **Hoe Tung Yew** [1,*], **Aroland Kiring** [1], **Seng Kheau Chung** [1], **Ali Farzamnia** [1], **Ali Chekima** [1] **and Manas Kumar Haldar** [2]

1 Faculty of Engineering, Universiti Malaysia Sabah, Kota Kinabalu 88400, Malaysia; mark_irwin_goh_mk20@iluv.ums.edu.my (M.I.G.)
2 Faculty of Engineering, Swinburne University of Technology Sarawak Campus, Kuching 93350, Malaysia
* Correspondence: htyew@ums.edu.my

**Abstract:** The evolution of 5G small cell networks has led to the advancement of vertical handover decision-making algorithms. A mobile terminal (MT) tends to move from one place to another and, as the 5G network coverage is small, user network access will change frequently and lead to a high probability of unnecessary handover, which is a waste of network resources and causes degradation of service quality. This paper aims to reduce the number of unnecessary handovers in 5G heterogeneous networks by proposing a handover decision-making algorithm that integrates the dwelling time prediction technique and Technique for Order of Preference by Similarity to Ideal Solution (TOPSIS). The proposed algorithm reduces the number of unnecessary handovers by estimating the connection time to a small cell network using the dwell time prediction technique. The TOPSIS evaluates the network quality and chooses the best network based on user preference. The result shows that the proposed handover algorithm reduces the number of unnecessary handovers to small cell networks in high-speed scenarios. It also saves the network connection cost by up to 27.51% compared with the TOPSIS-based handover algorithm. As for throughput achievement, the proposed algorithm yields an improvement of 5.12%. The proposed algorithm significantly reduces the number of unnecessary handovers in the high-speed scenario while fulfilling user preferences.

**Keywords:** vertical handover; TOPSIS; 5G Networks; unnecessary handover; dwelling time prediction





## 1. Introduction

The rapid development of wireless technologies has led to compatibility problems between the technologies that affect handover from one to another. The trend towards attaining pervasive access over heterogeneous technologies nowadays entail the addition of a variety of existing and upcoming networking technologies for a seamless wireless communication environment. Advances in wireless communication have led to multipurpose, convenient, and inexpensive network services. The interconnection between heterogeneous networks is a critical element that enables pervasive access. Achieving a seamless handover is a challenging issue. The evolution of next-generation networks is expected to consider handover management as its main aspect [1].

Handover management in a wireless network is a process that allows a mobile terminal (MT) to continue ongoing sessions when moving from one network access point to another. When the connection moves from one access point to another of similar technology, it is called "horizontal handover." On the other hand, a vertical handoff is a handover process between two distinct networks, such as handovers between a wireless local area network (WLAN) and a fifth-generation (5G) mobile network. Handovers can also be classified as hard and soft handovers. Hard handover is the type of handover that releases the serving station signal before the new resources can be committed. This type of handover causes interruption to mobile communication during the handover process [2]. Soft handover is

the type of handover where the mobile terminal establishes a mobile connection with a new access point before the old link is released. It performs a handover without interruption.

The handover process consists of three stages, as shown in Figure 1. The handover initiation stage detects networks and the main attributes of the network via the Media Independent Handover Function (MIHF) [3,4]. The handover decision making chooses the best network based on network information and initiates the handover at the appropriate time. Handover execution establishes the connection with the selected network and releases the old network. Several handover criteria can be used to perform a handover decision, such as received signal strength (RSS), data rate, monetary cost, bit error rate (BER), and signal-to-interference plus noise ratio (SINR). Using multiple criteria in performing handover decisions can provide a better handover performance than conventional handover algorithms that depend on one or two handover parameters.

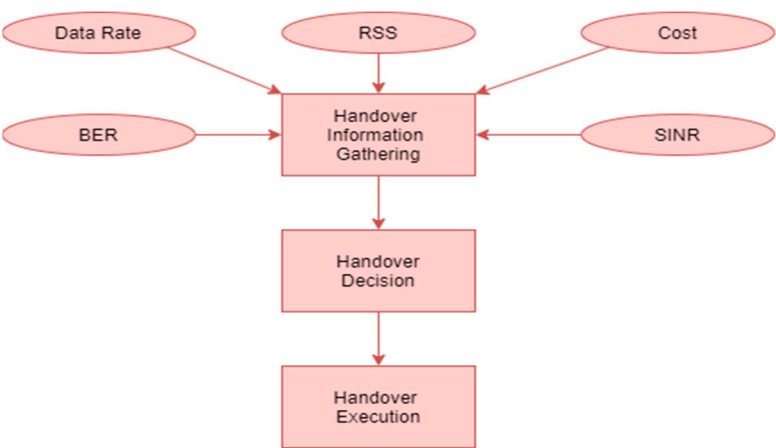

**Figure 1.** Handover process.

Fifth-generation network is the latest mobile wireless technology and is highly preferred because of its high data transmission rate and low latency [5]. However, the 5G network cell coverage is much smaller than its predecessor, the 4G network. The MT moving at high speed will suffer frequent handovers and lead to a high number of unnecessary handovers. This may cause data corruption, delay in transmitting data, and poor user experience in the network [6]. Therefore, an accurate handover decision-making algorithm is needed for achieving seamless handover.

The Technique for Order of Preference by Similarity to Ideal Solution (TOPSIS) is a reliable multi-attribute decision-making (MADM) technique and has been widely applied for handover decision making in heterogeneous networks. However, most MADM handover algorithms are biased towards large cell networks, such as 4G networks, in high-moving speed scenarios. The intelligent-based handover algorithm is more complex as it requires a data training process for algorithms to learn [7]. This paper presents a handover algorithm that integrates a dwell time prediction technique with the TOPSIS method to reduce the number of unnecessary handovers in 5G heterogeneous networks. The proposed algorithm avoids unnecessary handover by estimating the connection time to a 5G small cell network using the dwell time prediction technique, while the network evaluation and selection are performed by the TOPSIS algorithm [6].

## 2. Literature Review

A MADM algorithm uses various handover parameters, such as monetary costs, network security, available bandwidth, handover latency, and power consumption, for handover decision making [8,9]. The most commonly used MADM algorithms are TOPSIS and simple additive weighting (SAW). Each of them works in a different way, but they all select networks by comparing different options based on multiple attributes and choosing

the network that best satisfies the conditions. According to Lozano-Garzon et al. [10], the results of each are almost the same.

An adaptive vertical handover algorithm proposed by Li [8] improves user satisfaction and power consumption using a moving average method to predict future data and reduce the measurement rate in the network discovery phase. This method requires the latest data sets for data prediction. The network discovery is only triggered to collect the network's parameters, RSS and signal delay, and the user's parameters, including network cost, security, and power consumption, when the value of the cost function uptrends. Then the cost function for each discovered network is calculated. The network with the lowest cost will be selected as the handover target. The cost function is given by Equation (1).

$$Cost = w_C \log(C) + w_S \log(1/S) + w_P \log(P) + w_{RSS} \log(1/RSS) + w_D \log(D) \quad (1)$$

where $Cost$ is the cost function of the network; $C$ is the network cost for the user; $S$ refers to the security level of the networks (higher level better network security); $P$ represents power consumption; $RSS$ is the received signal strength of the network; $D$ is the delay of the network; and $w_C$, $w_S$, $w_p$, $w_{RSS}$, and $w_D$ are the respective weights of the above parameters, and their total weightage is 1, as shown in Equation (2).

$$w_C + w_S + w_P + w_{RSS} + w_D = 1 \quad (2)$$

The prediction of the cost function considers the user's preference. If network cost was set at the highest priority, the network with lower cost would be selected as an optimal network. This algorithm successfully reduces power consumption and cost. However, the algorithm might have a high number of unnecessary handovers when the MT traverses the small cell networks at high speed because the cost function did not take MT velocity into consideration.

TOPSIS-based MADM handover algorithms have been presented in [9,11–14]. Abdullah and Zukarnain [9] proposed an enhanced TOPSIS-based vertical handover algorithm for a heterogeneous network that consists of microwave access (WiMAX), WLAN, and LTE. The algorithm focuses on three user preferences: Gold Cost, Silver Cost, and Bronze Cost, as described in Table 1.

**Table 1.** User Preference algorithm [9].

| User Preference | Description |
| --- | --- |
| Gold Cost | Has the best quality of service (QoS) but it does not consider the cost of the network |
| Silver Cost | Balance in cost and QoS |
| Bronze Cost | Cost is considered more important than the QoS parameter |

The authors manually set the parameters' weightage for different user preferences. The handover parameters used were RSS and network usage. The algorithm reduces the number of handovers and the overall cost by up to 60% and 40%, respectively, as compared with the traditional RSS-based handover algorithm.

Goutam et al. [11] proposed a TOPSIS-based handover algorithm and tested it using a real environment 4G network and WLAN data. The input parameter for the TOPSIS algorithm included network coverage, packet loss, jitter, latency, bandwidth, and RSS. The testing results showed that the algorithm mostly selected the 4G network because the 4G network has better network quality than WLAN. This work was only tested in a low-speed environment by entering a café for the WLAN connection. Furthermore, the quality of WLAN heavily depends on the internet plan subscribed to by the café.

Yew et al. [12] proposed an improved TOPSIS-based handover algorithm in heterogeneous networks. The TOPSIS approach was used because of its multi-attribute decision analysis method, which simultaneously estimates the best and worst alternatives. The

algorithm was designed for telecardiology applications. During the critical condition where the patient is in emergency status, the algorithm is biased towards high-quality networks to maintain the QoS at the highest level. For non-critical situations, the algorithm is biased towards the low-cost network. The algorithm improves user satisfaction levels in terms of cost and QoS. The main drawback of this handover algorithm is that it will induce a high number of handovers if it continually searches for and connects to the small cell networks that offer better QoS. Additionally, user mobility profile, such as velocity, was not considered. Therefore, it is not suitable for performing handovers in a high-speed scenario.

Malathy and Muthuswamy [13] proposed using TOPSIS with a knapsack approach for handover between WiMAX and WLAN networks. The algorithm is focused on giving the best user satisfaction in terms of QoS. This was enabled by the knapsack approach, where the QoS Traffic parameter was calculated for the TOPSIS algorithm. The algorithm successfully reduces the handover failures and improves network performance compared with a traditional approach. However, they only tested in a low-speed environment, which is below 5 m/s. The number of handovers would increase while the MT moves at high speed.

Mathonsi et al. [14] proposed an intelligent intersystem handover algorithm that integrated the grey prediction theory (GPT), TOPSIS, fuzzy analytic hierarchy process (FAHP), and principal component analysis (PCA). The GPT is used to predict future RSS values, and FAHP calculates the assigned parameter weightage based on user preference. PCA evaluates the network QoS and selects the best network. The algorithm provides better network throughput. Kaur and Mittal [15] proposed a FAHP-based MADM handover algorithm that evaluates the priority vectors to decide when the handover should be initiated. The handover criteria considered were the signal-to-noise interference ratio, speed and MT moving direction. The algorithm successfully reduces the number of unnecessary handovers.

Intelligent handover decision-making schemes have varying degrees of complexity and intelligence. They have the advantage of handling complex and large parameters [16]. The intelligent-based handover decision-making algorithms have better handover performance but are more complicated and have higher handover latency than the MADM algorithm [17]. Furthermore, the unnecessary handover rate in [14,15] will increase proportionally with the increase of MT speed unless the algorithm is biased to large cell networks during the high-speed scenario. In such a case, it sacrifices the connection to small cell networks (5G), which provides a high QoS.

MADM handover scheme that is integrated with the neural network has been proposed by Xiaonan et al. [7] for heterogeneous wireless networks that involved WLAN, 4G, and 5G. The parameters involved were user data speed, Max and Min transmission rate, SINR, bit error rate (BER), and packet loss rate (PLR). The algorithm predicts the download rate of the available networks and the network with the highest download rate will be selected as a handover target. The handover success rate of this scheme is up to 90%, which shows efficient handover and seamless connectivity between the networks. However, the experiment was tested at a low moving speed only and the problem of unnecessary handover was not considered by the authors. Furthermore, this algorithm is more complicated, as it needs to conduct training and learning processes.

Yew et al. [18] proposed a UMTS-WLAN handover algorithm based on dwell time prediction, RSS, signal-to-noise ratio (SNR), and monetary costs. The dwell time, $t$, is predicted for WLAN using Equation (3), where $l$ is the estimated travelling distance across WLAN coverage and $v$ is the MT moving speed. The algorithm initiates dwelling time prediction technique when the MT detects the $RSS_{boundary}$ of WLAN, as shown in Figure 2. $R$ is the radius of the predefined boundary of WLAN coverage $RSS_{boundary}$ and $r$ represents the radius of the predefined RSS threshold $RSS_{th}$. The $R$ and $r$ values are computed using the log-distance path loss model. Available WLANs are considered candidates for handover

when their corresponding $t$ is higher than the time threshold (2 s). The unnecessary handover occurs if $t$ is less than 2 s.

$$t = \frac{l}{v} = \frac{R^2 - r^2 - d}{vd} \qquad \text{where } l = \frac{R^2 - r^2 - d}{d} \qquad (3)$$

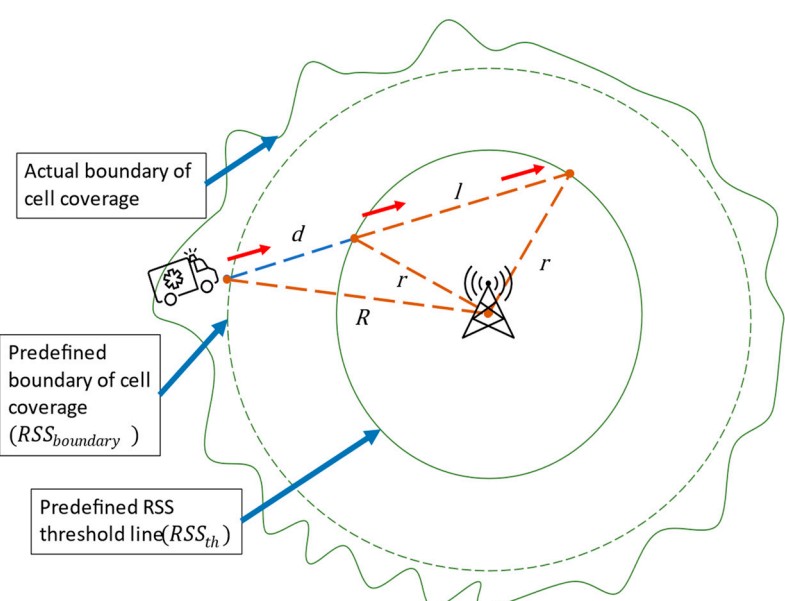

**Figure 2.** Prediction method to estimate travelling distance ($l$).

The quality of candidate, $k$ ($Q_k$), is evaluated based on the measured $RSS_k$, $SNR_k$, and monetary cost, $C_k$, values by using Equation (4), where the network with the highest $Q$ value is selected as the best network.

$$Q_k = \frac{RSS_k * (SNR_k - SNR_{REQ\_WLAN})}{C_k}, \ k = \{1, 2, \ldots, n\} \qquad (4)$$

The simulation results show that the proposed algorithm has better throughput compared with the conventional handover scheme. Furthermore, the algorithm minimizes the probability of unnecessary handover while optimizing the connection time to WLAN networks. However, the network quality evaluation is based on RSS and SNR only. More QoS criteria such as bandwidth, data rate, delay, and bit error rate (BER) should be considered for a full network quality assessment.

## 3. Methodology

This paper proposes a handover scheme that combines the dwelling time prediction technique and TOPSIS. The proposed method is less complicated than the intelligent-based handover algorithm because it does not require data training and learning, such as in the case of Xiaonan et al. [7]. The pseudocode of the proposed handover decision-making algorithm is shown in Figure 3. The MT initiates the handover process if the RSS of the serving network falls below the threshold value where it does not meet the user's service requirements.

```
1. DO
2. Monitor the quality of the serving network.
3.
4.      IF the serving network meet the user requirements THEN
5.          Back to step 2
6.      ELSE
7.          Scan for alternate networks
8.      END IF
9.
10.     IF a small cell network detected THEN
11.         Apply dwelling time prediction technique to estimate t in the small cell network
12.             IF t ≥ 2 seconds THEN
13.                 Evaluate the quality of the network using TOPSIS method
14.             ELSE
15.                 Reject the small cell network
16.             END IF
17.     ELSE
18.         Evaluate the quality of the network using TOPSIS method
19.     END IF
20. Rank all available network candidates based on the TOPSIS score.
21. Initiate handover to the best network.
22. WHILE (1)
```

**Figure 3.** Pseudocode of the proposed handover algorithm.

The proposed algorithm will trigger the dwelling time prediction algorithm if any small cell network such as 5G or WLAN is detected [18]. The dwell time, $t$, within the small cell network can be computed using the method presented in [18] by dividing the estimated travelling distance, $l$, in the small cell network with MT speed, $v$, in m/s, as expressed by Equation (3). The MT speed, $v$, can be obtained through the vehicle speedometer. The dwelling time prediction algorithm only allows the small cell networks with an estimated connection time greater than two seconds to undergo network quality evaluation using the TOPSIS method. The TOPSIS method was selected because it can handle multi-criteria parameters and complex information in the handover process [11]. The network candidate with the highest score was to be selected as the targeted network. The handover parameters used by the proposed algorithm for handover decision making include RSS, throughput, cost, delay, bit error rate, and signal-to-noise plus interference ratio (SNIR). Each parameter was assumed to take monotonically increasing values and the positive ideal solution would consist of the best values of all parameters attained by all the alternatives. Meanwhile, if each parameter took monotonically decreasing values, the negative ideal solution would comprise the worst parameter values attainable from all alternatives.

The TOPSIS network quality evaluation process is expressed as follows [19]:

1.   Decision matrix construction, $a$

$$
a = 
\begin{array}{c}
\begin{matrix} P_1 & \dots & P_j & \dots & P_n \end{matrix} \\
\begin{bmatrix}
a_{11} & \dots & a_{1j} & \dots & a_{1n} \\
\vdots & & \vdots & & \vdots \\
a_{i1} & \dots & a_{ij} & \dots & a_{in} \\
\vdots & & \vdots & & \vdots \\
a_{m1} & \dots & a_{mj} & \dots & a_{mn}
\end{bmatrix}
\begin{matrix} N_1 \\ \vdots \\ N_i \\ \vdots \\ N_m \end{matrix}
\end{array}
\tag{5}
$$

where $N_i$ is the $i$th network candidate, $P_j$ is the $j$th of type of handover parameter, $a_{ij}$ represents the value of network parameter $i$ with respect to $j$, $m$ is the maximum of row in the matrix, and lastly, $n$ is maximum of column in the matrix.

2.   Normalize the decision matrix using the normalization method expressed in Equation (6) and form the normalized decision matrix $A$ as Equation (7).

$$A_{ij} = \frac{a_{ij}}{\sqrt{\sum_{j=1}^{n} a^2{}_{ij}}} \quad for\ benefit\ attribute,\ \ i = 1, 2, \ldots m, \quad j = 1, 2, \ldots n \quad (6)$$

$$A = \begin{bmatrix} A_{11} & \cdots & A_{1j} & \cdots & A_{1n} \\ \vdots & & \vdots & & \vdots \\ A_{i1} & \cdots & A_{ij} & \cdots & A_{in} \\ \vdots & & \vdots & & \vdots \\ A_{m1} & \cdots & A_{mj} & \cdots & A_{mn} \end{bmatrix} \quad (7)$$

where the $A_{ij}$ is the value of normalization from $a_{ij}$.

3.　The weighted normalized decision matrix, $X$, is given as

$$X = A * W = \begin{bmatrix} w_1 A_{11} & \cdots & w_j A_{1j} & \cdots & w_n A_{1n} \\ \vdots & \vdots & \vdots & \vdots & \vdots \\ w_1 A_{i1} & \cdots & w_j A_{ij} & \cdots & w_n A_{in} \\ \vdots & \vdots & \vdots & \vdots & \vdots \\ w_1 A_{m1} & \cdots & w_j A_{mj} & \cdots & w_n A_{mn} \end{bmatrix} = \begin{bmatrix} X_{11} & \cdots & X_{1j} & \cdots & X_{1n} \\ \vdots & \vdots & \vdots & \vdots & \vdots \\ X_{i1} & \cdots & X_{ij} & \cdots & X_{in} \\ \vdots & \vdots & \vdots & \vdots & \vdots \\ X_{m1} & \cdots & X_{mj} & \cdots & X_{mn} \end{bmatrix} \quad (8)$$

where $W$ is the handover parameter weightage allocation matrix as expressed in Equation (9).

$$W = \begin{bmatrix} w_1 & 0 & 0 & 0 & 0 & 0 \\ 0 & w_2 & 0 & 0 & 0 & 0 \\ 0 & 0 & \ddots & 0 & 0 & 0 \\ 0 & 0 & 0 & w_j & 0 & 0 \\ 0 & 0 & 0 & 0 & \ddots & 0 \\ 0 & 0 & 0 & 0 & 0 & 0 \\ 0 & 0 & 0 & 0 & 0 & w_n \end{bmatrix} \quad (9)$$

The total weightage of all handover parameters must be equal to 1, as expressed in Equation (10).

$$\sum_{j=1}^{n} w_j = 1 \quad (10)$$

4.　Finding the ideal solutions (positive, $I_+$, and negative, $I_-$). The ideal solutions for each handover parameter are as expressed in Equations (11) and (12).

$$I_+ = \max\left(X_{ij} \middle| j \in n\right) \quad (11)$$

$$I_- = \min\left(X_{ij} \middle| j \in n\right) \quad (12)$$

where $I_+$ represents the best value of the value among the parameters of the network and $I_-$ is the opposite of $I_+$.

5.　Calculate the distance of the ideal solutions between $I_+$ and $I_-$ points. The distance from point $I_+$ to $I_-$ is represented by $S_{+_i}$ and $S_{-_i}$, respectively, as expressed by Equations (13) and (14).

$$S_{+_i} = \sqrt{\sum_{j=1}^{n} \left(X_{ij} - I_{+_i}\right)^2} \quad (13)$$

$$S_{-_i} = \sqrt{\sum_{j=1}^{n} \left(X_{ij} - I_{-_i}\right)^2} \quad (14)$$

6.　The relative closeness, $Q_i$, of each available network is calculated by using Equation (15). The network candidate with the highest score is the best network and was to be selected as the handover target.

$$Q_i = \frac{S_{-i}}{S_{-i} + S_{+i}} \tag{15}$$

Without integrating the dwelling time prediction technique with the TOPSIS, MT is unable to measure the MT dwelling time in the small cell network. It might lead to a high number of unnecessary handovers because the TOPSIS will select the best network regardless of connection time. It does not benefit the user if the dwell time is too short; it is a waste of network resources. In this work, we assumed MT to cross the small cell network in a straight line.

## 4. Experimental Setup

The proposed handover is simulated by 100 loops in the heterogeneous wireless network environment that consists of 4G LTE, 5G, and WLAN networks, as shown in Figure 4. The 4G LTE has the most comprehensive network coverage, covering all the 5G and WLAN networks. This assumes that the MT initially connects to the 4G LTE and moves from point A to destination B traverses the heterogeneous networks at high speed (40–100 km/h). The simulation parameters are tabulated in Table 2 [7,12].

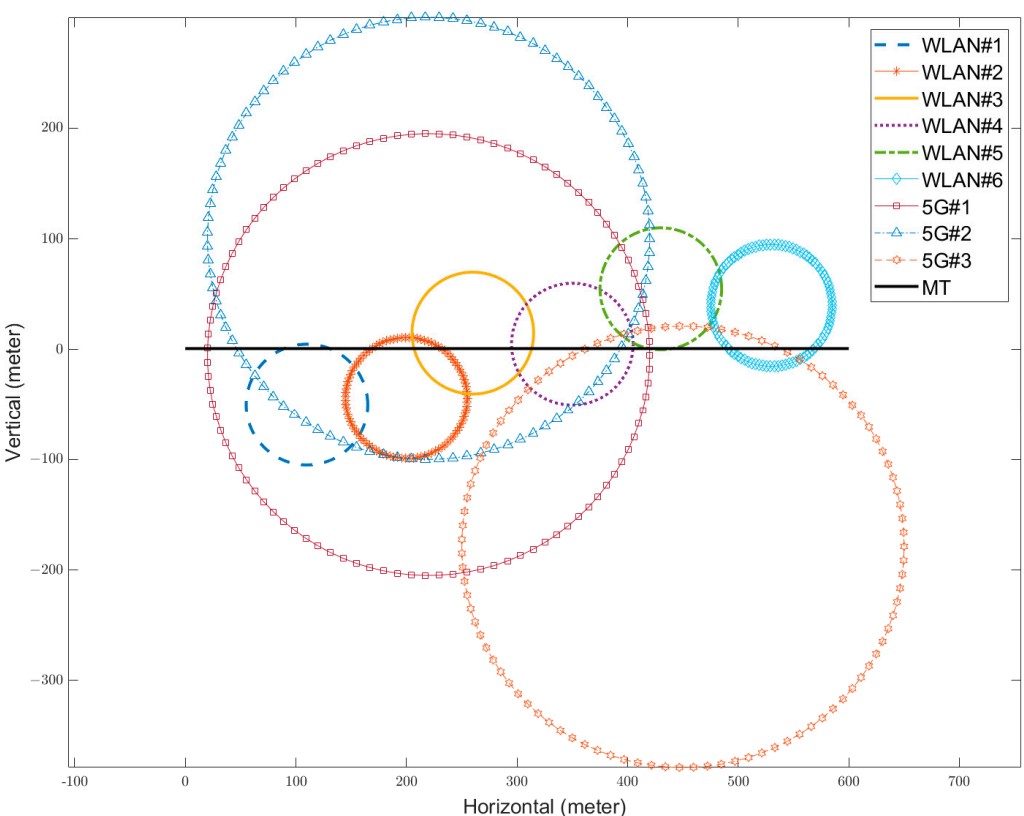

**Figure 4.** Experimental setup for the heterogenous wireless network's environment.

The user usually prefers the WLAN due to its high bandwidth and low cost. However, the network coverage of WLAN is very limited. Therefore, many studies have excluded WLAN from the high-speed scenario and have prioritized a 4G network that has large network coverage [20]. In the new era, the 5G network will be a future trend for wireless communication because it provides the best network quality among wireless network candidates. However, the 5G network coverage is up to 300 m only [21]. In this work, the proposed handover algorithm monitors the network quality at the time interval of $z$. The time interval is adjusted automatically according to the MT velocity, given as

$$z = \frac{D}{v} \tag{16}$$

where $D$ is 1 m and $v$ is the MT velocity that can be obtained through the vehicle speedometer. We are assuming the MT only collects data and passes them to the network orchestration for handover processing. Therefore, the proposed algorithm will not overload the MT.

**Table 2.** Simulation parameters [7,12].

| Parameters | 5G | WLAN | LTE |
|---|---|---|---|
| Carrier frequency (GHz) | 28 | 2.4 | 1.8 |
| RSS threshold (dBm) | −90.6 | −91 | −92 |
| Radius (m) | 200 | 55 | 2000 |
| Unit cost per Mbps (assumed) | 50 | 1 | 8 |
| Throughput (Mb/s) | 1000 | 50 | 20 |
| Delay (s) | 1 | 7 | 15 |
| Bit error rate (BER) (%) | 0.01 | 7 | 0.08 |
| SINR | 25 | 22 | 20 |
| PLR | 1 | 20 | 10 |
| Handover cost | 2 units per handover | | |
| MT speed (km/h) | 40 to 100 | | |
| Adaptive time interval, T (s) | $1/v$ | | |
| Dwell time threshold (s) | 2 | | |
| Health services U (Mb) | 0.827 Mb = ECG + vital + audio + video | | |
| ECG (Mb) | 0.024 | | |
| Vital (Mb) | 0.010 | | |
| Audio (Mb) | 0.025 | | |
| Video (Mb) | 0.768 | | |

The proposed handover algorithm takes the mean value of RSS samples to minimize the oscillation of RSS signals. The RSS values, $RSS_m$ can be obtained using Equation (17) [22]. The formula for calculating the RSS mean value, $\overline{RSS}$, is given in Equation (18).

$$RSS_m = P_{TX} - PL_0 - 10nlog\frac{R}{d_0} + \varepsilon \tag{17}$$

$$\overline{RSS} = \frac{\sum_{i=1}^{o} RSS_m}{q} \tag{18}$$

where $q$ is the number of RSS values taken, $P_{TX}$ is the transmit power, $PL_0$ is the power loss at the reference point, $n$ is the path loss exponent, $d_0$ is the distance between the AP and a reference point, and $\varepsilon$ is a zero-mean Gaussian random variable caused by shadow fading. The difference between RSS signals found by taking a single RSS sample and the mean value of multiple RSS samples is shown in Figure 5. The RSS signal using the mean value is smoother than the RSS signals shown in Figure 6. This could minimize the ping-pong effect in heterogeneous networks.

The distance threshold, $l_{th}$, is the product of the MT velocity ($v$) and dwelling time threshold. The two seconds dwelling time threshold for different MT velocities yields different travelling distance thresholds, as indicated in Figure 7. The $l_{th}$ is increased proportionally with the MT velocity. Table 3 displays the trajectory distance within each network based on the network layout in Figure 4. The predicted distance is the distance calculated by the dwelling time algorithm. The Table 3 results for WLAN#1 and 5G#3 have more than 10% error due to the short actual travel distance when less than 20 m. However, the two networks' actual travel distances and prediction travel distances have approximately values.

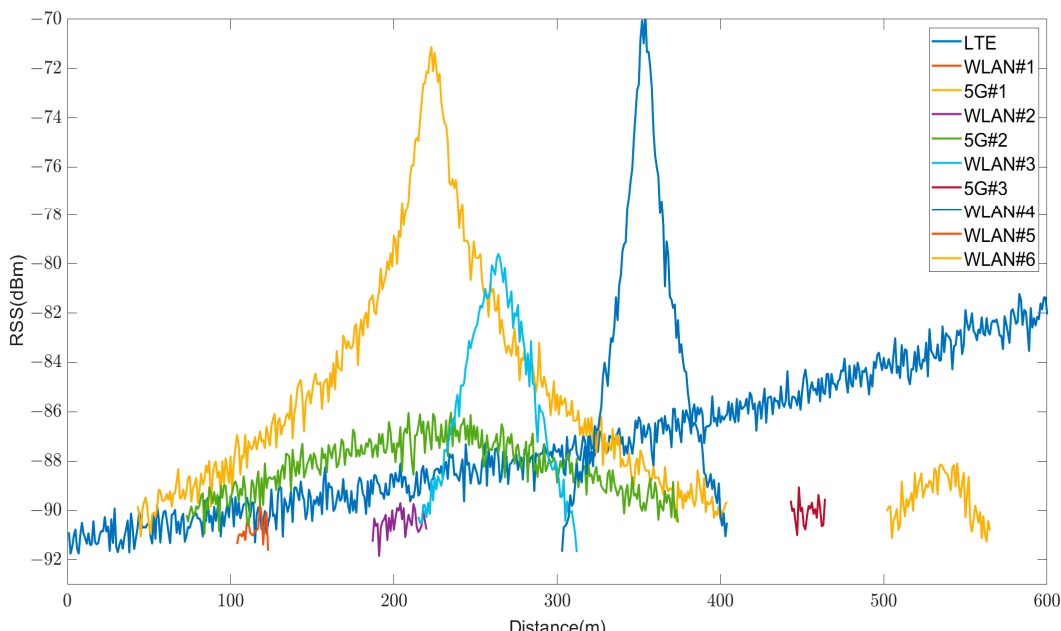

**Figure 5.** RSS signal by taking a single RSS sample.

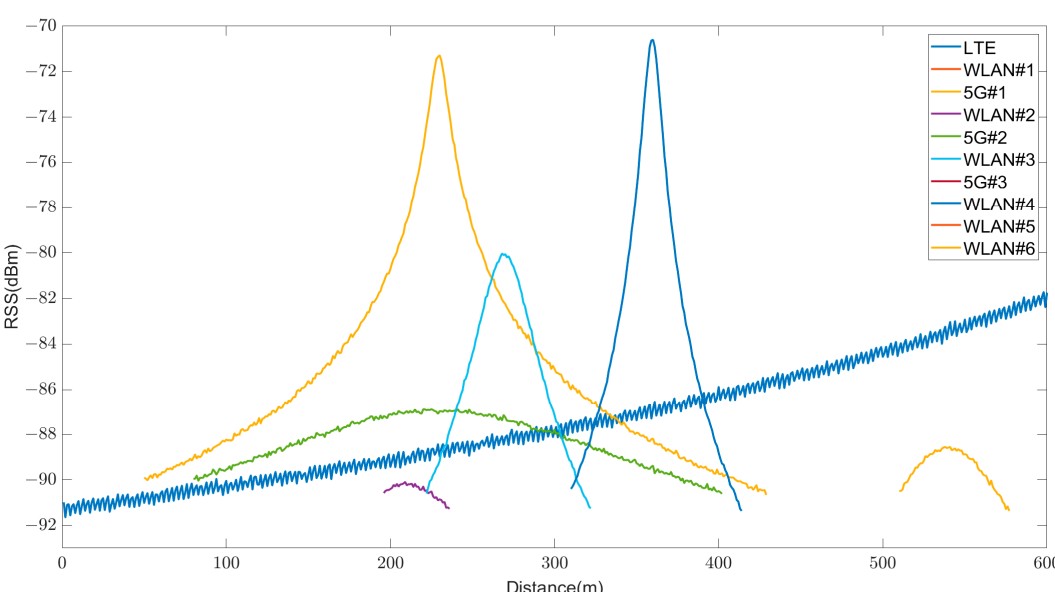

**Figure 6.** RSS signal by taking the mean RSS value.

**Table 3.** Trajectory in each network.

| WLAN Access Point /5G Base Station | Actual Distance (m) | Predicted Distance (m) | Error (%) |
|---|---|---|---|
| WLAN#1 | 10 | 13 | 30 |
| WLAN#2 | 30 | 33 | 10 |
| WLAN#3 | 106 | 97 | 8.49 |
| WLAN#4 | 110 | 101 | 8.18 |
| WLAN#5 | 0 | 0 | 0 |
| WLAN#6 | 90 | 86 | 4.44 |
| 5G#1 | 393 | 379 | 3.59 |
| 5G#2 | 300 | 310 | 3.33 |
| 5G#3 | 18 | 16 | 11.11 |

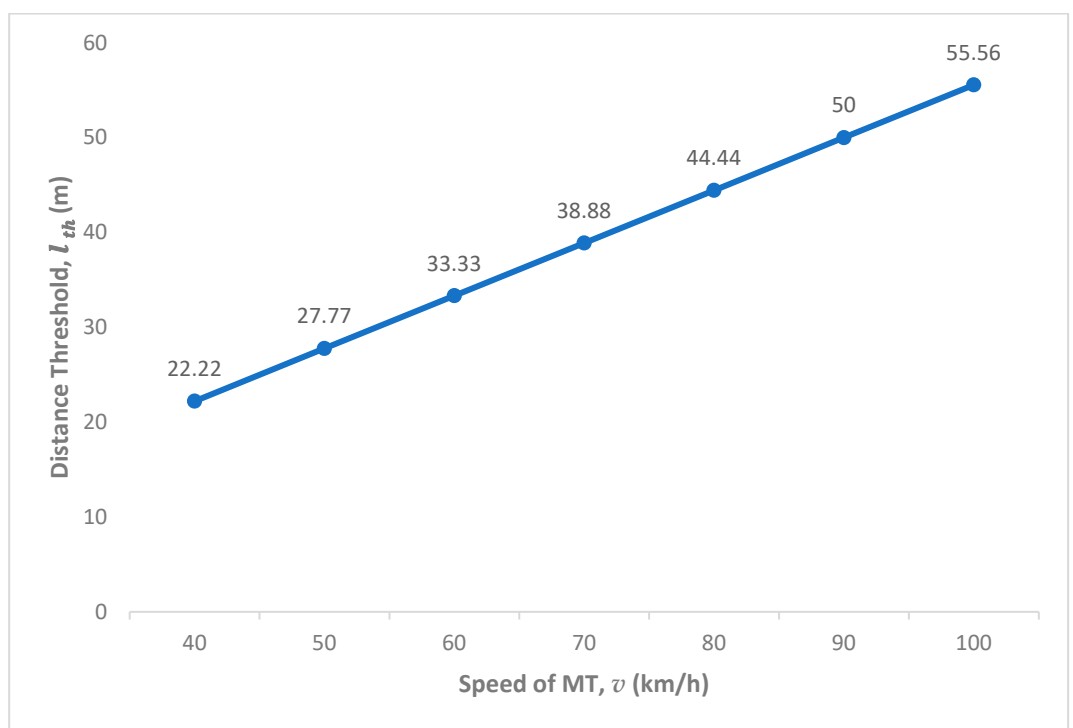

**Figure 7.** Distance threshold ($l_{th}$) for different speeds.

In this work, the TOPSIS weight allocation is set based on two user preferences, cost, and QoS. The users who select the cost preference aim to minimize the network connection cost while maintaining the service quality, whereas the QoS preference is for those who strive for the best service quality with no care about the network connection cost. The weight allocations for the QoS preference and cost preference are shown in Table 4. For cost preference, higher weightage is assigned to the parameter cost. On the other hand, QoS preference is given higher weightage on the QoS parameters such as data rate, delay, BER, and SINR. This is assuming the MT is transmitting health data (ECG + vital + audio + video) in real time [23]. The TOPSIS scoring for each network under different user preferences is indicated in Table 5. Each network scoring value is computed using Equation (15) based on the weightage allocation shown in Table 4. The higher the score, the better the network.

**Table 4.** TOPSIS weightage allocation for cost and QoS preferences.

| Parameter | Cost | Data Rate | Delay | BER | SINR | PLR |
|---|---|---|---|---|---|---|
| Cost preference | 0.95 | 0.01 | 0.01 | 0.01 | 0.01 | 0.01 |
| QoS preference | 0.1 | 0.3 | 0.15 | 0.15 | 0.15 | 0.15 |

**Table 5.** TOPSIS scoring for each network under different user preferences.

| Network | LTE | WLAN | 5G |
|---|---|---|---|
| Cost preference | 0.975 | 0.977 | 0.951 |
| QoS preference | 0.358 | 0.376 | 0.991 |

## 5. Results and Discussion

The performance of the proposed handover algorithm is here benchmarked against the TOPSIS-based handover algorithm [11]. The main difference between the two algorithms is that the TOPSIS-based handover algorithm has no dwell time prediction technique. It always selects the network with the highest score. As shown in Table 5, the cost preference

prioritizes the low network connection cost of WLAN, followed by LTE and 5G networks. In this case, the TOPSIS-based handover algorithm will select WLAN whenever it is available. Figures 8 and 9 show the handover performed by the TOPSIS-based handover algorithm and the proposed handover algorithm at the speed of 40 km/h and 70 km/h, respectively. The TOPSIS-based handover algorithm induces a number of unnecessary handovers as it is biased toward WLAN even though the connection time to WLAN is very limited and does not contribute any benefit to the MT. In contrast, the proposed method has no unnecessary handovers because the dwelling time prediction technique implemented in the proposed handover algorithm rejects the network with an estimated dwelling time of fewer than two seconds.

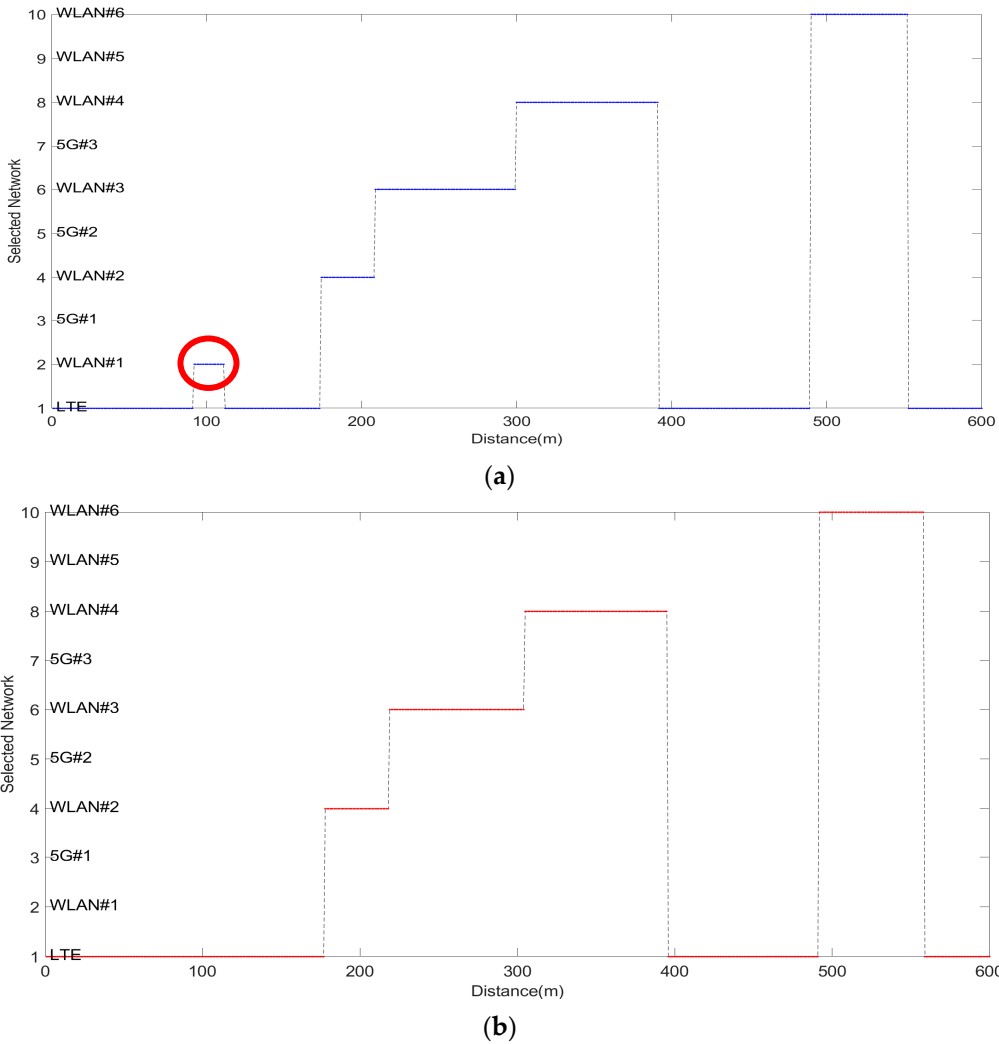

**Figure 8.** Handover performed based on cost preference at the speed of 40km/h: (**a**) TOPSIS-based handover (circle shows unnecessary handover) and (**b**) proposed handover algorithm.

The handover to WLAN#1 is considered an unnecessary handover (circled in red in Figure 8a) at the speed of 40 km/h and above. Referring to Figure 7, WLAN#1 has a trajectory of 10 m, and the dwelling time at the speed of 40 km/h was 0.9 s which is less than the predefined threshold of two seconds. With the dwelling time prediction technique, the proposed handover algorithm avoids unnecessary handover to WLAN#1, as shown in Figure 8b. It rejects all the networks with an estimated dwelling time less than two seconds. At 70 km/h and above, the dwelling time for WLAN#1 and WLAN#2 drops below two seconds. Figure 9b shows that the proposed handover algorithm rejected these networks.

However, the TOPSIS-based handover algorithm performed unnecessary handover to these networks (Figure 9a).

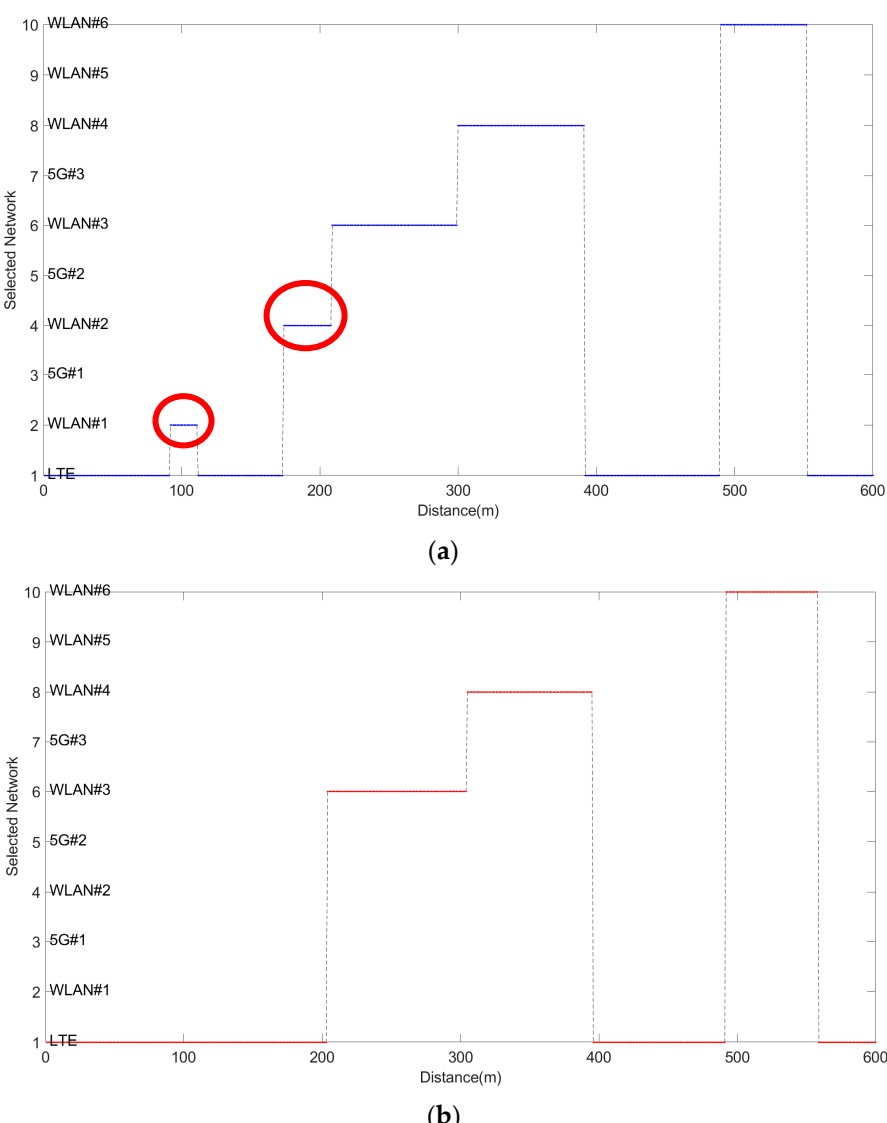

**Figure 9.** Handover performed based on cost preference at the speed of 70 km/h (**a**) TOPSIS-based handover (circles show unnecessary handover) and (**b**) proposed handover algorithm.

The handover performed by the TOPSIS-based handover algorithm and proposed handover algorithm for the QoS preference at 40 km/h and 70 km/h is shown in Figures 10 and 11, respectively. In this case, the 5G network is highly preferred. As shown in Table 5, the 5G network scores the highest points under QoS preference. At the speed of 40 km/h, an MT should avoid connection to 5G#3 because the trajectory in 5G#3 coverage is short and is 18 m, as indicated in Table 3. The travelling time, *t*, equals 1.62 s, which is less than two seconds. However, the TOPSIS-based handover algorithm performed handover to 5G#3. The unnecessary handovers are marked with a red circle, as shown in Figure 10a. A similar result for the speed of 70 km/h, the TOPSIS-based handover algorithm is connected to the 5G network whenever it is detected where the unnecessary handover occurs at 5G#3.

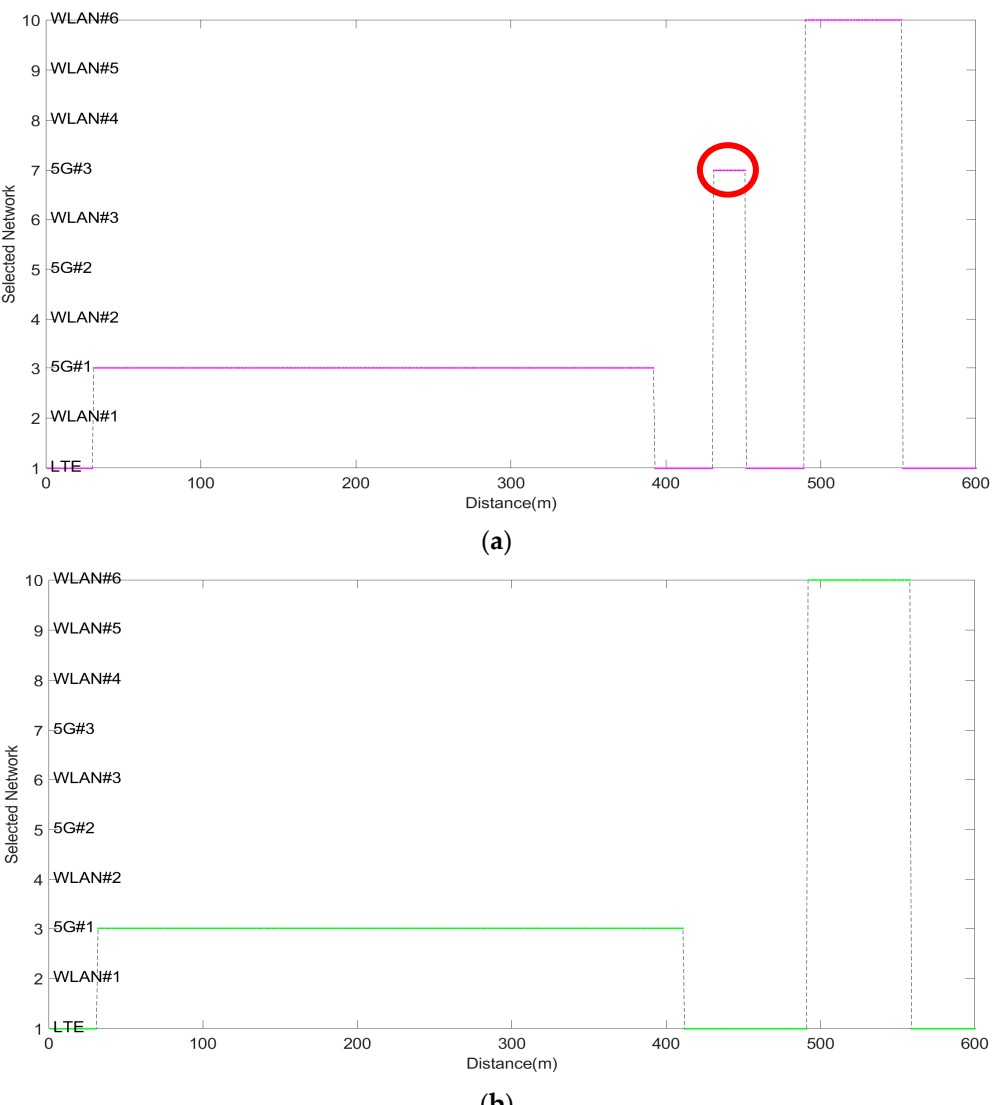

**Figure 10.** Handover performed based on QoS preference at the speed of 40 km/h: (**a**) TOPSIS-based handover algorithm (circle shows unnecessary handover) and (**b**) proposed handover algorithm.

The performance of the proposed method is also compared with the TOPSIS-based handover algorithm in terms of total connection cost and throughput. Assuming the MT moves from left to right in the scenario, as shown in Figure 5, the $i$th network dwelling time, $t_i$, can be calculated using Equation (19).

$$t_i = \frac{l_i}{v} \tag{19}$$

where $l_i$ is the trajectory distance in the $i$th network. The $i$th network connection cost, $C_i$, can be determined by

$$C_i = t_i \times C_{iM} \times U + C_{handover} \tag{20}$$

where $C_{iM}$ is the $i$th network cost per Mb, $U$ is the service data rate requirement in Mb and $C_{handover}$ is the handover cost. The total connection cost ($C_{total}$) is the sum of all types of network connection costs. It is given as

$$C_{total} = C_{LTE} + C_{WLAN} + C_{5G} \tag{21}$$

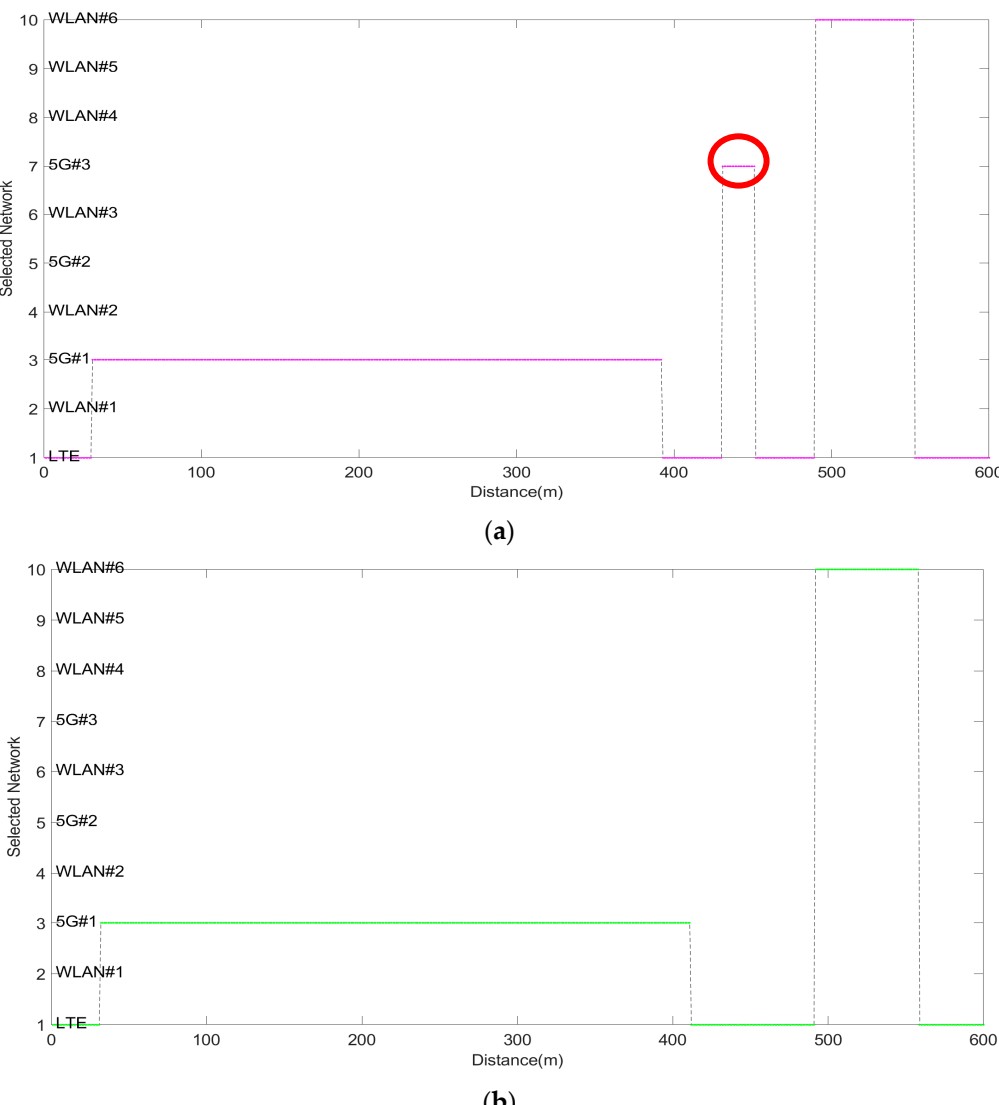

**Figure 11.** Handover performed based on QoS preference at the speed of 70 km/h: (**a**) TOPSIS-based handover algorithm (circle shows unnecessary handover), (**b**) proposed handover algorithm.

The throughput in Mb (*Tput*) can be computed based on network dwelling time, $t_i$, the *i*th network data rate in Mbps, $N_i$, and loss of the network data rate for each unnecessary handover, $T_{iloss}$, as expressed in Equation (22). $T_{iloss}$ is expressed in Equation (23), where $t_{iunnecessary\ ho}$ is unnecessary handover time. The total throughput ($Tput_{total}$) is the sum of the throughput for all types of access points, as expressed in Equation (24).

$$Tput_i = t_i \times N_i - T_{iloss} \tag{22}$$

$$T_{iloss} = N_i \times t_{iunnecessary\ ho} \tag{23}$$

$$Tput_{total} = Tput_{LTE} + Tput_{WLAN} + Tput_{5G} \tag{24}$$

The network connection period is inversely proportional to MT speed. The higher the MT speed, the shorter the network connection period. Therefore, the total connection cost and throughput reduce while the MT moves at higher speeds. Figure 12a,b show the total connection cost and throughput obtained by the MT when it opted for cost preference, respectively.

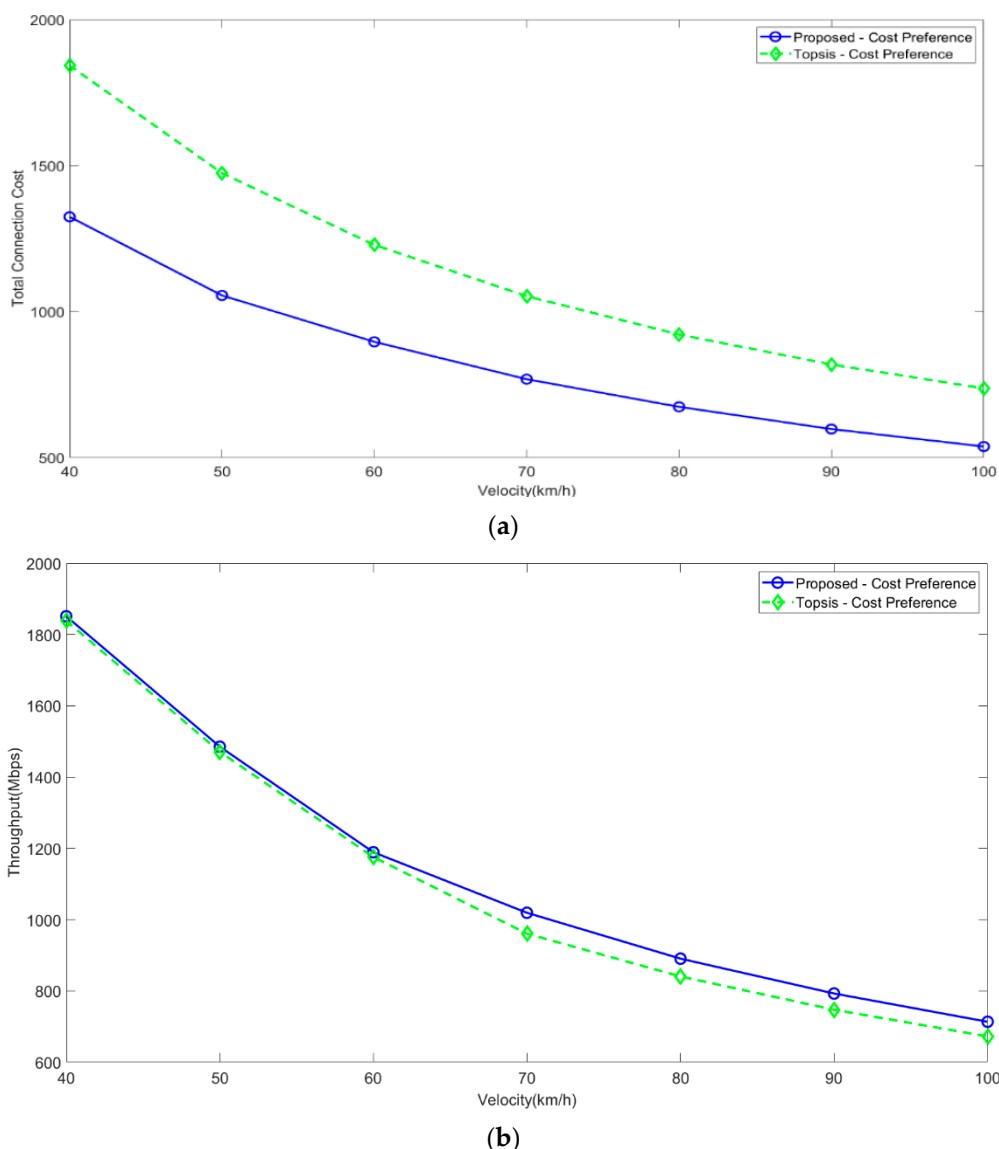

**Figure 12.** Performance of cost preference in terms of (**a**) total connection cost and (**b**) throughput.

Referring to Table 5, the algorithms give the highest priority to WLAN, which has the lowest network connection cost, and then continue to LTE and 5G. Figure 12a shows that the proposed algorithm reduces the total connection cost up to 27.51% compared with the TOPSIS-based handover algorithm. In terms of throughput, the proposed algorithm is 3.06% higher than the TOPSIS-based handover algorithm, as shown in Figure 12b. At the lower speed (40 to 60 km/h), there is a small throughput difference between the two algorithms, as fewer unnecessary handovers occur at low speeds. More unnecessary handovers induced at higher speeds (70 to 100 km/h) caused the throughput to decrease significantly for the TOPSIS-based handover algorithm. The unnecessary handovers are a waste of network resources.

Figure 13 shows the total connection cost and throughput achieved by the proposed handover algorithm and TOPSIS-based handover algorithm based on the QoS preference. In this case, the MT is biased toward the 5G network, which offers high service quality. The TOPSIS-based handover algorithm connects to the 5G network wherever it is available. Figure 13a shows that the total connection cost of the proposed algorithm is 40.81% lower than that of the TOPSIS-based handover algorithm. In terms of throughput, the proposed algorithm is 5.12% better than the TOPSIS-based handover algorithm. For example, the

unnecessary handover to network 5G#3 (as shown in Figures 10a and 11a) performed by the TOPSIS-based handover algorithm is a waste of cost and network resources.

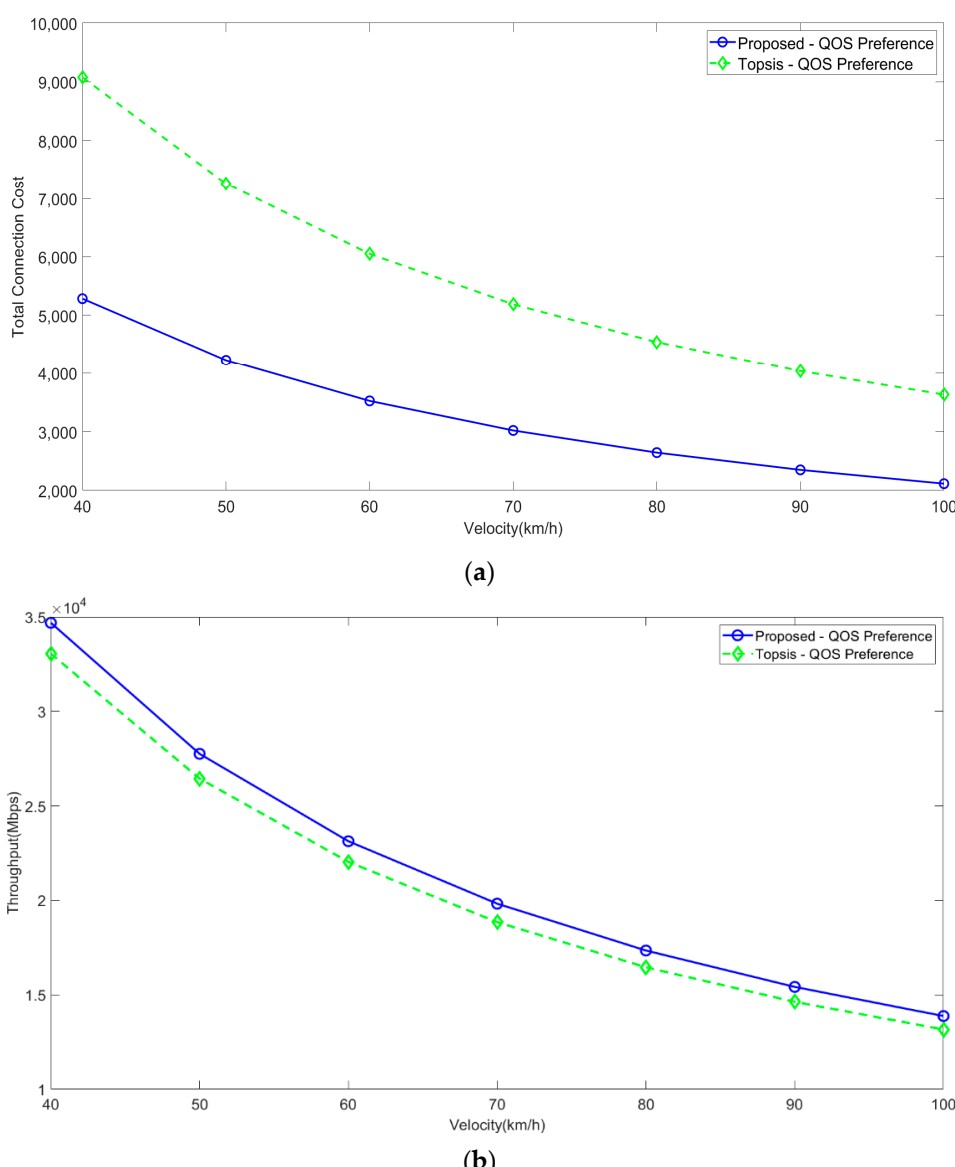

**Figure 13.** Performance of QoS preference in terms of (**a**) total connection cost and (**b**) throughput.

The total connection cost and throughput of the QoS-preference-based handover are much higher than the cost-preference-based handover because the 5G network cost is more expensive and offers better service quality. If no wireless network meets the user requirements, the proposed algorithm connects to the best network among the available networks based on user preference. The adaptive network selection mechanism [24] can be applied to overcome this issue.

Figure 14 shows the number of handovers performed by the proposed algorithm and TOPSIS-based handover algorithm for both the cost preference and QoS Preference in a 100 loops simulation. The number of handovers for the TOPSIS algorithm on both preferences is higher than that of the proposed algorithm. Table 6 summarizes the performance of the prosed algorithm as compared with the TOPSIS-based handover algorithm.

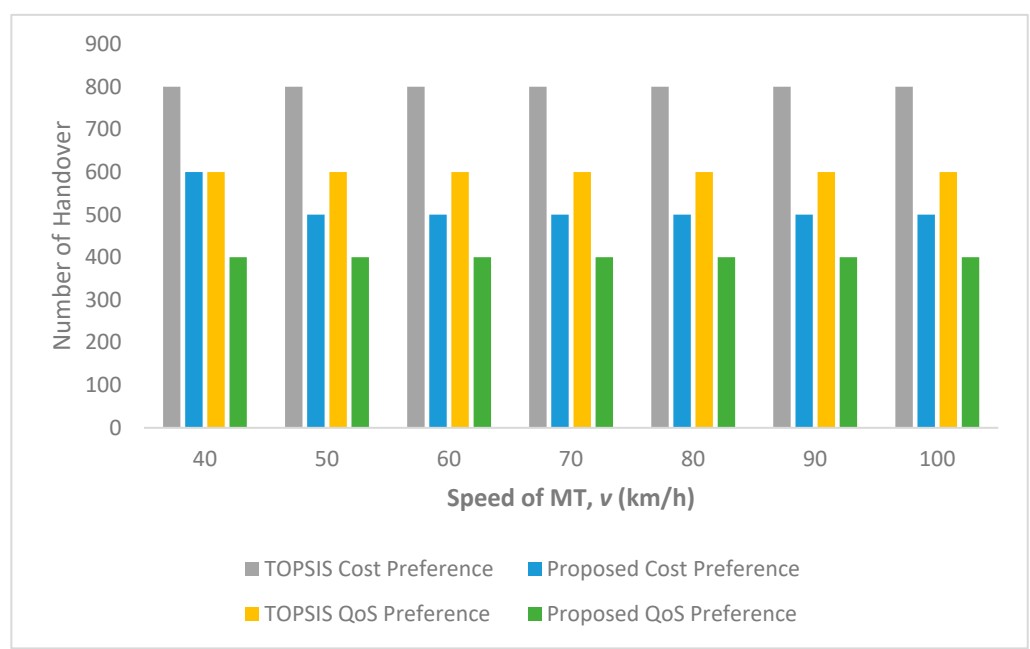

**Figure 14.** Number of handovers and unnecessary handovers for 100 loop simulation.

**Table 6.** Performance of the proposed algorithm.

| Parameter | Compared with TOPSIS-Based Handover Algorithm [11] |
|---|---|
| Network connection cost | −27.51% |
| Throughput | +5.12% |
| Number of handovers for cost preference based | −35.71% |
| Number of handovers for QoS preference based | −33.33% |

## 6. Conclusions

This paper has proposed a handover algorithm that integrates the dwell time prediction technique and the TOPSIS method. The handover decision-making process takes MT speed, network cost, and various QoS parameters into consideration. Heterogeneous wireless networks consisting of LTE, WLAN, and 5G networks have been simulated. The results show that the proposed algorithm significantly reduces the number of handovers in the high-speed scenario while satisfying user preferences for cost preference and QoS preference up to 35.71% and 33.33%, respectively. The proposed handover algorithm outperforms the TOPSIS-based handover algorithm in terms of cost and throughput as well. It reduces the network connection cost and improves the throughput by up to 27.51% and 5.12%, respectively. The proposed algorithm could improve user experience in the new era of heterogeneous networks.

This research focuses on a high-speed environment with the assumption of an MT crossing the small cell network in a straight line. The proposed handover algorithm can be further improved and tested in an ultra-dense scenario where the MT moves in random directions.

**Author Contributions:** Conceptualization, H.T.Y., A.K. and S.K.C.; methodology, M.I.G., A.I.M. and H.T.Y.; software, M.I.G., A.I.M. and H.T.Y.; validation, M.I.G., A.I.M., H.T.Y., M.K.H., A.F. and A.K.; formal analysis, M.I.G., A.I.M., H.T.Y., M.K.H., A.F. and A.C.; investigation, M.I.G., A.I.M. and H.T.Y.; resources, H.T.Y., A.K. and S.K.C.; writing—original draft preparation, M.I.G., A.I.M. and H.T.Y., writing—review and editing, H.T.Y., M.K.H., A.F. and A.C.; supervision, H.T.Y., A.K. and S.K.C.; project administration, H.T.Y. and S.K.C.; funding acquisition, H.T.Y. All authors have read and agreed to the published version of the manuscript.

**Funding:** This research was funded by the Ministry of Higher Education Malaysia, Fundamental Research Grant Scheme (FRGS) FRGS/1/2020/TK0/UMS/02/2. The APC was funded by Research Management Centre, Univerisiti Malaysia Sabah.

**Acknowledgments:** The authors wish to express their appreciation to the Ministry of Higher Education Malaysia, Fundamental Research Grant Scheme, and Universiti Malaysia Sabah funded the research.

**Conflicts of Interest:** The authors declare no conflict of interest.

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
