# Peer review of "Handover Decision-Making Algorithm for 5G Heterogeneous Networks"

_electronics, doi:10.3390/electronics12112384_

Round 1
Reviewer 1 Report
This paper proposed a handover algorithm that integrates the dwell time prediction technique and TOPSIS method. This research work in this field belongs to a more concerned research direction, many scholars and communication companies are concerned about this issue. Especially for 5G and 6G communication, it has important engineering application value.
But before the article can be considered for publication, there are a number of issues that need to be addressed.
1、 The core innovation of this paper is to propose a new algorithm,which is integrated by the dwell time prediction technique and TOPSIS method. The feasibility analysis that these two algorithms can integrate should be given in detail for the Methodology part. What problems are encountered after integration? What is the theoretical basis of integration?
2、 The introduction part needs to give out to aim at this problem, what algorithm has been used in the existing research? What are the problems? Is the algorithm used in this paper adopted?
3、 In the body part of Experimental Setup, the verification of the new algorithm is insufficient. To verify the validity and superiority of a new algorithm, it is necessary to verify its accuracy, speed and range of application. Not only need to put in the chart and data, but also according to the chart information for analysis, and finally give a clear conclusion.
4、 There is also much room for improvement in the accuracy and rigor of the words used in the article writing.
Author Response
Point 1: The core innovation of this paper is to propose a new algorithm,which is integrated by the dwell time prediction technique and TOPSIS method. The feasibility analysis that these two algorithms can integrate should be given in detail for the Methodology part. What problems are encountered after integration? What is the theoretical basis of integration?
Response 1: Integration are further added in page 8 line 257-263. This is to explain the reasons of the integration. The limitation encountered in the experiment to test in random directions, this paper experiment only simulated straight line direction. The basis of the integration is avoiding unnecessary handover by the TOPSIS algorithm.
Point 2: The introduction part needs to give out to aim at this problem, what algorithm has been used in the existing research? What are the problems? Is the algorithm used in this paper adopted?
Response 2: Existing research TOPSIS algorithm [11] is used. Problems are most MADM handover algorithms such as TOPSIS algorithms are biased towards large cell networks like 4G network in high-moving speed scenarios, stated in introduction page 2 line 76 -86.
Point 3: In the body part of Experimental Setup, the verification of the new algorithm is insufficient. To verify the validity and superiority of a new algorithm, it is necessary to verify its accuracy, speed and range of application. Not only need to put in the chart and data, but also according to the chart information for analysis, and finally give a clear conclusion.
Response 3: Added Table 6 to show the improvement percentage of the algorithm in Results and Discussion page 18 Line 426.
Point 4: There is also much room for improvement in the accuracy and rigor of the words used in the article writing.
Response 4: Improved.
Reviewer 2 Report
In this work, the aurthors proposed algorithm which has significantly reduced the number of unnecessary handovers in the high-speed scenario while fulfilling user preferences. Few of the suggestions are:
1. The figures quality throughout the manuscript is low. it should be improved.
2. It would be better to add a comparison table at the end of the manuscript to compare the key performance parameters achieved in this work with the literature work.
3. As the authors algorithm has improved the handover. So it would be nice to provide some experimental results based on the proposed algorithm which would verfiy the performance of the proposed algorithm further.
4. As the handover improvement is for 5G networks. So usually, sub-6 GHz and mm-wave bands are under consideration for the 5G. It would be better to highlight that the proposed algorithm is suitable in both chunks of bands transmission or only in one case?
Author Response
Point 1: The figures quality throughout the manuscript is low. it should be improved.
Response 1: Figure text font increased and quality pixel increased.
Point 2: It would be better to add a comparison table at the end of the manuscript to compare the key performance parameters achieved in this work with the literature work.
Response 2: TOPSIS algorithm is adopted from [11]. Figure 14 added to show the performance of 100 loop simulation of the algorithms in page 18 Line 417.
Point 3: As the authors algorithm has improved the handover. So it would be nice to provide some experimental results based on the proposed algorithm which would verify the performance of the proposed algorithm further.
Response 3: Table 6 added to show the improvement done by proposed algorithm page 18 Line 426.
Point 4: As the handover improvement is for 5G networks. So usually, sub-6 GHz and mm-wave bands are under consideration for the 5G. It would be better to highlight that the proposed algorithm is suitable in both chunks of bands transmission or only in one case?
mm-wave bands are used. However, the prediction algorithm can be used either one regardless the band due to the algorithm has been used in 4G as well from existing research. Therefore, the prediction method can be used any network and main point using it is on small cell network due to short travelling distance.
Reviewer 3 Report
The manuscript aims to propose a handover algorithm that integrates the dwell time prediction technique and the TOPSIS method. It considers MT speed, network cost, and various QoS parameters. The simulation was run on wireless networks such as LTE, WLAN, and 5G. The authors claimed that the proposed algorithm has significantly reduced the number of unnecessary handover numbers and outperforms the TOPSIS-based handover algorithm in terms of cost and throughput. I believe that the manuscript is suitable for publication in the journal "Electronics" after addressing these minor issues.
Minor points:
1. Please rewrite the paragraph (lines 92-103); the whole paragraph is copied from a source. https://ieeexplore.ieee.org/document/8393221
2. Provide limitations of the experiment and a way forward in a subsection.
3. Are the figures and equations necessary to put in the literature survey? A literature survey is to get to know the existing research in the same area and the limitations of those research, some of which the authors will try to solve.
4. What is the rationale behind pseudocode line 12 (the value 2)?
5. Does the TOPSIS network quality evaluation process – author's novel contribution? If not, they need to provide citations from where they got the idea of the evaluation process. (also for all equations ).
6. Table 3 is copied, so use citations and do not copy/paste.
Author Response
Point 1: Please rewrite the paragraph (lines 92-103); the whole paragraph is copied from a source. https://ieeexplore.ieee.org/document/8393221
Response 1: Rewrote.
Point 2: Provide limitations of the experiment and a way forward in a subsection.
Response 2: Added in Conclusion body page 18 Line 439-442.
Point 3: Are the figures and equations necessary to put in the literature survey? A literature survey is to get to know the existing research in the same area and the limitations of those research, some of which the authors will try to solve.
Response 3: Previous Figure 3 has been removed. Added [11] in literature Review body page 3 Line 124 -130.
Point 4: What is the rationale behind pseudocode line 12 (the value 2)?
Response 4: The value 2 is 2 seconds. It is the handover time threshold. If predicted dwell time, t, is less than 2 seconds, the proposed method will reject the network candidate to avoid unnecessary handover.
Point 5: Does the TOPSIS network quality evaluation process – author's novel contribution? If not, they need to provide citations from where they got the idea of the evaluation process. (also for all equations ).
Response 5: Cited [20].
Point 6: Table 3 is copied, so use citations and do not copy/paste.
Response 6: Table 3 changed to Figure 7 and Table 4 changed to Table 3 edited with percentage error.
Round 2
Reviewer 2 Report
The revision seems good.